# Case Study: An Evaluation of Detection Dog Generalization to a Large Quantity of an Unknown Explosive in the Field

**DOI:** 10.3390/ani11051341

**Published:** 2021-05-08

**Authors:** Edgar O. Aviles-Rosa, Gordon McGuinness, Nathaniel J. Hall

**Affiliations:** 1Department of Animal and Food Sciences, Texas Tech University, Lubbock, TX 79409-214, USA; edgar.aviles-rosa@ttu.edu; 2Independent Researcher, Vancouver, BC V3E3B9, Canada; gord1168@shaw.ca

**Keywords:** detection dog, concentration generalization, ammonium nitrate fuel oil, canine olfaction, explosive

## Abstract

**Simple Summary:**

This case study is a series of experiments to investigate a real-life event where two highly trained and certified detection dogs did not find an explosive in a suspicious bag. We tested seven dog teams from the agency in three experiments and confirmed that dogs were able to detect the agency’s training sample in a small quantity (30 g) but not the large amount of the confiscated explosive (13 kg) found in a similar scenario. To further evaluate a possible generalization deficit, we tested dogs with a 30 g subsample of the confiscated explosive, and most of the dogs were able to detect it (but with some decrement) even though they largely failed to detect 13 kg of the same material. Finally, we trained dogs to detect the 30 g subsample until reaching proficiency and found that after training with the small sample, dogs still showed poor generalization to the large-quantity sample until explicit training with the large sample was conducted. Altogether, this series of studies shows the importance of explicitly training for larger quantity finds and has led to changes in agency practices to mitigate future risks.

**Abstract:**

Two explosive detection dogs were deployed to search a suspicious bag, and failed to detect 13 kg of explosive within. The aim of this research was to further evaluate this incident. First, dog teams (*N* = 7) searched four bags in a similar scenario. One bag contained the same 13 kg of explosive, two bags were blanks, and the other contained the training sample that the agency routinely used for training. All dogs detected the training sample, but most (5/7) did not alert to the 13 kg sample. Subsequently, dogs received two trials in a line up with a 30 g subsample of the explosive to evaluate whether they could generalize to a smaller quantity. Most dogs (6/7) alerted to the subsample at least once. Finally, dogs were trained with the 30 g subsample and later tested with the 13 kg sample. Only three dogs spontaneously generalized to the large sample after training with the small subsample. Dogs’ alert rate to the 13 kg sample was improved with training in subsequent trials with the 13 kg sample. This result indicates that explosive detection dogs may not generalize to a target odor at a significantly higher quantity relative to the one used in training, highlighting the importance of conducting such training.

## 1. Introduction

Despite the technological advances in chemical detection techniques, dogs continue to be widely used as field detectors for drugs, explosives, agricultural pests, and even human diseases [1,2,3,4,5,6,7,8]. To effectively use dogs as detectors, it is important to understand cognitive and behavioral phenomena that can influence performance such as stimulus generalization and discrimination. Stimulus generalization is when an animal responds to stimuli differing from the trained discriminative stimulus [9]. Generalization typically occurs when the discriminative stimulus and the new stimulus share some common properties [9]. Stimulus discrimination occurs when a conditioned behavior is under strict stimulus control and the animal will not respond to stimuli that differ in some manner from the trained stimulus [9,10]. It is important to note that both phenomena occur in all sensory modalities, but we focus our discussion on olfaction as it is the most relevant to detection dogs.

The success of dogs as detectors relies on their sensitivity and ability to generalize and discriminate between similar and dissimilar odors. Olfactory generalization refers to the phenomenon where a dog produces the same response to perceptually similar odors that could share the same outcome [11,12]. In the field, olfactory generalization occurs when a dog alerts or shows the same trained behavioral response to variations of a target odor [13,14]. Olfactory discrimination is the opposite phenomenon where dogs do not alert to variations of the trained odor [11,14]. Both olfactory phenomena have a significant impact on detection dogs’ performance and in their ability to detect and alert to variations of a target odor they may encounter in the field.

Explosive and narcotic detection dogs need a balance between generalization and discrimination. For instance, methyl benzoate is thought to be the main volatile present in street cocaine samples [7]. However, this same compound is present in the snapdragon flower [7,14]. Thus, a narcotic detection dog should be able to generalize between different sources of street cocaine that could vary in the amount of methyl benzoate present, but at the same time they must discriminate and not alert to other non-illegal objects that contain methyl benzoate, such as snapdragon flowers. Olfactory generalization is of extreme importance to dogs trained to detect improvised explosive devices (IED), becuase these explosives are often built with different types of oxidizers (fertilizer, ammonium nitrate, potassium chloride, etc.), fuels (petroleum jelly, gasoline, diesel, etc.), and additional materials. Thus, the IED a dog will encounter in the operational environment will most likely be different in at least some respect from the ones used in training [15].

Despite the importance of olfactory generalization and discrimination in detection dogs, to date, there is limited research on this topic. Furthermore, the current research suggests that dogs tend to be very specific to the explosive with which they were trained and tend to show low generalization to variations of the same explosive. For instance, 87% of Labrador retrievers trained to detect potassium chlorate failed to generalize to explosive mixtures containing potassium chlorate [16]. Similarly, less than 64% of the dogs trained to detect laboratory grade ammonium nitrate (AN) responded to fertilizer grade AN or AN mixtures at a rate greater than chance [15]. To increase generalization, some trainers use multiple variations of the target odorant during training. However, recent studies have found that increasing the number of AN variants used in training did not improve generalization to other variants or to a mixture of AN and aluminum flakes [12,17].

Further, generalization is important when dogs need to detect odor mixtures. Dogs trained with a mixture of two odorants (AB) generalized better to a mixture containing a novel odorant than dogs trained to detect each odorant individually [18]. Similarly, training with different mixtures containing AN or hydrogen peroxide increased generalization to novel mixtures containing these oxidizers [19].

The concentration of the odorant used in training is also an important dimension to consider, because a recent study found that dogs will generalize spontaneously to a change in concentration not greater than 10 fold [13]. Further, explicitly training dogs for lower concentrations leads to improvements in odor detection thresholds [20]. Lastly, when concentrations of an odorant change substantially, the perceptual quality (at least for humans) of the odorant can also change substantially [21].

In this case study, we conducted a series of experiments following a scenario where two certified explosive detection dogs did not alert to a suspicious bag containing 13 kg of a putative ammonium nitrate fuel oil (ANFO) mixture. Two of the seven explosive detection dogs of the agency were deployed to investigate a suspicious bag abandoned in an open field. After sniffing the bag, neither of the canines indicated that the bag contained an explosive. It was later determined that the bag contained 13 kg of ANFO with a hydrophilic thickener. The aim of this study was to conduct a series of rapid tests to evaluate why these certified dogs failed to alert to this large quantity of explosives to (1) remedy the potential security threat and (2) communicate this finding broadly to prevent future incidents. We hypothesized that dogs failed to alert for the following reasons: (1) the context was unusual, and performance may not have generalized to a single bag in an open-field search; (2) the specific putative ANFO substance was too different from the ANFO training samples, causing a generalization failure; (3) dogs did not generalize due to the quantity (and presumably odor concentration), which was substantially larger than what the agency was permitted to train with. A brief window of opportunistic testing was made available with the confiscated material and data collected and reported here.

## 2. Methods and Results

### 2.1. General

Seven detection dog teams were used in this study. A dog team consisted of a certified explosive detection dog and their handler (Table 1). Detection teams were regularly trained with 30 g of an ammonium nitrate fuel oil (ANFO) sample as part of maintenance training. Larger quantities were rarely trained or available due to safety and policy constraints. All dogs met all the training requirements to be deployed as explosive detection dogs at the time of testing. This series of experiments was conducted at an undisclosed location where the explosive device was maintained and under investigation. All studies were conducted in a single day in successive order for logistical reasons.

Data were collected by members of the agency and subsequently sent to the Canine Olfaction Lab at Texas Tech University for analysis and interpretation. Data collection was determined by the Texas Tech Institutional Animal Care and Use Committee to be opportunistic and did not include researcher involvement outside of observing and recording normal training and was therefore considered an observational study, not requiring an animal care and use protocol. All statistical analyses were conducted using R and SAS statistical software [22,23]. Binomial tests and Wilson’s score 95% confidence intervals were calculated using PROC FREQ and the generalized linear model used in Experiment 2 was estimated using PROC GLIMMIX in SAS. The Cochran test was calculated using the cochrans.q functions from the nonpar library of R studio. A significant difference was declared when *p* < 0.05. The raw data and a descriptioncan be found in the Appendix A of this manuscript (Appendix A).

### 2.2. Experiment 1

#### 2.2.1. Methods

Experiment 1 was conducted to (1) confirm the scenario in question was a real phenomenon of dog failure (i.e., not a chance event) and (2) evaluate whether the two dogs failed to alert to the large quantity of an unknown ANFO sample because of inexperience with the context (single bag in an open environment) and not due to olfactory differences of the target material. To evaluate this possibility, all seven detection dog teams were tested in a similar scenario to the one in question.

Four bags, similar to the one of the incident, were placed, isolated from each other, in an open field. Bag 1 contained the 13 kg putative ANFO mixture confiscated from the scenario in question. Bags 2 and 3 were blank bags stuffed with newspaper to make it look as if the bags were full. This was to prevent the dogs from getting any visual cues from the empty bags. Bag 4 was stuffed as bags 2 and 3 but it also contained 30 g of the ANFO mixture typically used during training. This bag served as a positive control as it contained the training sample dogs were familiar with. Failure to alert to bag 4 would suggest that inexperience with the context might have had a negative impact detecting the 13 kg of explosive. Alternatively, failure to detect the large-quantity sample could have been due to a lack of olfactory generalization. This would be suggested if the canine teams alerted to bag 4 (control) but not to the large-quantity sample at a rate greater than what might be expected by chance. The position and the order at which each detection team sampled bags 1, 2, and 3 was randomized such that each dog experienced each bag in a different order. The last search for all teams was bag 4 (control) to not influence dogs’ performance in the initial bag search of the large-quantity sample.

Dog handlers were blind to the treatment and which or whether any bags contained the explosive. A team approached the lead trainer who then instructed the team to go in a specific direction to search a bag in an open field and report upon return whether the dog alerted or not. The trained alert response for all teams was to sit and stare at the target for approximately 3 s.

After returning, the team would then be instructed which bag to search next until all 4 bags were searched. Each bag was in a distinct location and only one bag was present in the immediate vicinity. This was done to mimic the situation in question as much as possible. Correct responses were not reinforced as we were evaluating the dogs’ ability to generalize between samples (training vs. unknown) in this unusual context. The experimenter recorded whether a dog alerted or not to each bag.

The Cochran test was used to determine whether dogs alerted to each bag at a different rate. This model included dog team as a block and the fixed effect of bag [24]. Statistical significance was considered when *p* < 0.05.

#### 2.2.2. Results

Figure 1 shows the average performance of all seven detection teams. The effect of bag on dogs’ alert rate was statistically significant (Cochran’s Q = 14.66; df = 3; *p* = 0.002). All teams detected the bag containing the training sample (alert rate = 100%). Only two of the seven dog teams alerted to the bag containing the 13 kg confiscated sample (alert rate = 28.57%). One of the dogs that alerted to the confiscated sample was also one of the dogs deployed in the original incident. Notwithstanding, both dogs that alerted to the 13 kg sample, also alerted to one or both blank bags. Thus, it is possible that they alerted to the target sample by chance in this test. None of the remaining five dog teams alerted to any of the blank bags.

### 2.3. Experiment 2

#### 2.3.1. Methods

Results from Experiment 1 replicated dogs’ failure to detect the target, indicating that the incident in question was unlikely to be a chance occurrence/failure. Secondly, the results suggested that the context had no effect on dogs’ performance as all dogs readily alerted to the known training sample in the unusual context. Thus, a lack of olfactory generalization to the specific explosive in question seemed to be the main reason for why the two deployed dogs did not alert to the explosive.

In Experiment 2, we evaluated whether the lack of generalization to the 13 kg ANFO was due to the large quantity of explosive that this sample contained compared to what is normally trained. A generalization test was conducted using a five-choice line up. The line up consisted of 5 boxes, each one containing a different odor (or no odor). One box was blank (no odor), three boxes contained distractors (non-target) odors that dogs would commonly encounter, and the remaining box contained a 30 g subsample of the confiscated ANFO. This subsample was the target odor. Each team (*N* = 7) received two line up trials in which the dogs were walked along the line of scent boxes for the dog to evaluate. The handlers were blind to the position of the target sample and distractors, and the experimenter changed the position of all the boxes for each team and for each trial. The handler signaled the experimenter when a dog alerted or did not alert to any box, and the experimenter recorded the response as a correct, false alert or no response. Responses were not reinforced to assess spontaneous (non-reinforced) generalization.

After the second trial with the 30 g subsample, all teams received two additional trials in the same line up, but this time the target odor was 30 g of the ANFO sample trainers routinely used to train the dogs as a positive control test. Only a correct response in the second trial of the control test was reinforced. A correct response was defined as when a dog showed the trained behavioral response to the box containing the target odor.

Because the Cochran test used in Experiment 1 is not designed to assess the effect of multiple factors (factorial design), we used a generalized linear model with a binomial distribution to evaluate the effect of trial and sample (unknown vs. training) on the alert rate. The model included the fixed effect of trial, sample, and their interaction and a random effect of dog team. A binomial test was conducted to evaluate whether dogs’ alert rate to the 30 g subsample, and their alert rate to the training sample (Five-alternative choice test: H_0_
*p* = 0.20) were above chance in each trial.

#### 2.3.2. Results

Results of this experiment are presented in Figure 2. The effect of trial, sample, and their interaction were not statistically significant (all with *p* > 0.05). Four of the seven teams (57.14%) alerted to the 30 g subsample in both trials (binomial probability of a dog making two correct responses where probability of a correct response is 0.20, *p* = 0.04). Two dog teams alerted to the unknown sample in only one trial (team 1 in trial 1 and team 7 in trial 2). One of these teams was deployed in the original incident. The remaining dog team did not alert to the subsample in either of the trials and this team was the second team deployed in the original incident, but this dog did alert to the target bag in Experiment 1.

All dog teams alerted to the training sample in both trials (alert rate = 100%), indicating a strong response to the training sample. On average, dog teams alerted to the 30 g subsample of the confiscated explosive in 10 of 14 trials across all dogs at an alert rate of 71.43%. This suggests that most dogs were able to generalize to the confiscated ANFO when presented at a similar quantity used in training, albeit at a noticeably lower rate suggesting some generalization decrement due to the sample itself. This generalization decrement did not reach statistical significance.

### 2.4. Experiment 3

#### 2.4.1. Methods

Results from Experiment 2 showed that most dogs were able to generalize to the 30 g subsample of the confiscated ANFO, although with some decrement. This result suggests that the substantial lack of generalization observed in the real-life scenario could have been due to inexperience with high quantities of explosives. In Experiment 3, we further evaluated the effect of explosive concentration or quantity in olfactory generalization by evaluating if dogs will spontaneously generalize to the 13 kg confiscated sample after being trained to detect and alert to a 30 g subsample of the same 13 kg confiscated ANFO. Using the same five-choice line up, dogs were trained to detect 30 g of the confiscated ANFO sample. Dogs were trained following the agencies typical training procedure. Dogs were explicitly reinforced for correct responses. If a dog made a false alert, the handler simply waited until the dog moved from the false alert to the correct box and alerted. Training continued until dogs showed three consecutive correct runs, alerting on the 30 g of confiscated sample (binomial test, *p* = 0.008).

Subsequently, three bags were placed in a line up scenario to evaluate whether dogs were able to detect the 13 kg sample after training and demonstrating simple proficiency with a 30 g subsample. Two of the three bags were stuffed with paper (blanks) and the third bag contained the same 13 kg confiscated ANFO sample that the two deployed dogs failed to detect. Each detection dog team (*N* = 7) received three trials with the bags line up to evaluate whether they were able to generalize spontaneously to the large-quantity sample after been trained with a small quantity of the same sample. Handlers were blind to the test and the experimenter randomized the order of the bag between trials and teams. The handler indicated to the experimenter when a dog alerted, and the experimenter recorded dogs’ response as correct or incorrect. In this experiment, all correct responses were reinforced as we wanted to also evaluate whether detection of the large-quantity sample could be improved with training. Additionally, the agencies typical training procedures were used to facilitate dogs’ detection and remedy detection failures after the dogs’ initial response was recorded. If the dog made a false response on a trial, the trial was scored as incorrect, and the handler waited until the dog stopped alerting to the incorrect bag and continued the line up until the dog alerted at the correct bag and was reinforced. If the dog made no response, an incorrect response was scored, and the dog was brought back to the correct bag until making an alert and was reinforced.

A binomial test was used to evaluate whether dog teams alerted to the 13 kg sample at a rate greater than chance in each trial (H_0_*: p* = 0.33). An alert rate greater than chance in trial 1 suggests that dogs were able to spontaneously generalize to the 13 kg sample after training with 30 g of the same sample. The Cochran test was used to determine the effect of trial. Because correct responses were reinforced and incorrect responses corrected, we expected dogs’ alert rate to increase with trials. This would suggest that training to high quantities of explosive could resolve generalization problems.

#### 2.4.2. Results

Following training to proficiency (3 consecutively correct trials) with the 30 g subsample of the confiscated explosive, dogs’ alert rate (42.86%; 3 of 7 dogs) to the 13 kg sample was not significantly different from chance during the first trial (binomial test, 3 of 7 success with probability of a success of 33%, *p* = 0.58), suggesting that, overall, spontaneous generalization did not occur from the small quantity to the large quantity. Both dogs deployed in the original incident spontaneously generalized in trial 1. Alert rates to the 13 kg sample increased during trials 2 (71.43%; binomial test: 5 successful dogs of 7 dogs with the probability of success of 33%, *p* = 0.03) and 3 (85.71%; binomial test: 6 successful dogs of 7 dogs with the probability of success of 33%, *p* < 0.01 Figure 3). Only one dog did not alert to the confiscated sample in any of the three trials. Dogs that alerted during trial 1 also alerted in trials 2 and 3, and dogs whose initial response was in trial 2 also alerted in trial 3. The effect of trial on the alert rate trended toward statistical significance, but did not meet the 0.05 threshold (*p* = 0.10).

Altogether, Experiment 3 suggests that, after training dogs to proficiency with 30 g of the confiscated explosive, only 3 of 7 dogs generalized to 13 kg of the same explosive on the first trial. However, training dogs to alert to the large-quantity samples, could mitigate this generalization failure, where six of seven dogs responded to the 13 kg explosive after two training trials with the large quantity explosive.

## 3. Discussion

Previous studies in explosive detection dogs have found that search-related behaviors and detection responses could be extinguished in a context that dogs associate with low probability of encountering a target [25,26]. This association of a context with low probability of finding a target could reduce dogs’ search vigilance in an operational setting making them prone to miss a target when present [25,26]. Thus, our first hypothesis was that the two dogs did not alert to the unknown large sample because they had previously associated an operational scenario with low probability of finding a target odor. However, when tested in a scenario like the one in question, we found that all dogs were able to detect the 30 g training sample, but most of the dogs (five out of seven) failed to detect the 13 kg sample. These results suggested that the lack of response to the large quantity of explosive was likely not due to a context effect because all dogs alerted to the training sample in the same context where they failed to detect the unknown large sample. This indicates that dogs perceived the 13 kg sample as a non-target odor and thus did not alert.

In Experiment 2, we found that most dogs were able to generalize and alert to the 30 g subsample of the confiscated explosive. There was, however, a notable decrement in detection of the 30 g subsample (~72% alert rate: 5 of 7 dogs on the first trial and 10 alerts of 14 trials overall) compared to the training sample (100% alert rate). This does suggest that the explosive sample did have differences from the more familiar training sample. This could result from different ammonium nitrate sources, different fuel oil sources or differences in additives. The confiscated sample was later determined to include an unspecified hydrophilic thickener, which may in part have been responsible for this decrement from 100% alert rate to 72% alert rate.

Nonetheless, the substantially poorer alert rate in Experiment 1 (28%), suggests that the inability of the deployed dogs to detect the 13 kg explosive could have been due also to its large quantity, which would also agree with previous literature. For instance, using an air dilution olfactometer, DeChant et al. found that dogs spontaneously generalized to a change in concentration of only approximately 10 fold [13]. In our case, a 13 kg sample represented an increase of 433 fold relative to the quantity dogs were trained to detect. This extreme change in quantity most likely changed dogs’ perception of the ANFO. For instance, human participants reported the same odorants to be a different odorant if its concentration changed by 100 fold [21]. Humans also described the same odorant at different concentrations with different qualities [27]. Thus, we speculate that even when humans and dogs have significant differences in their olfactory system, the same perceptual phenomenon could be happening when dogs are presented with a large quantity of explosive. This possible perceptual difference could explain why dogs did not alert to the large sample in Experiment 1, as it may have been perceived as a different odor.

The results from Experiment 3 show that after training to proficiency (3 consecutive correct detections) with the small-quantity subsample, only 3 of 7 dogs (42.86%) generalized spontaneously to the 13 kg sample. This highlights that even with proficiency of 30 g of the exact same explosive material, only 3 of 7 dogs alerted to 13 kg of that material. This supports our hypothesis that the failure of detection that initiated this study was largely due to a failure to generalize to the 13 kg of the confiscated ANFO when only trained on 30 g of ANFO. Importantly, however, Experiment 2 does indicate generalization was also decreased by unfamiliarity with the specific sample of the explosive in question, although this decrease did not reach statistical significance. Nevertheless, our results also show that the dog teams’ response to the large-quantity sample was improved with brief training. This shows that the lack of generalization to large-quantity samples of an odorant can be overcome with training and access to large quantities for training.

It is important to note that the two dogs deployed in the original incident were able to spontaneously generalize to the large-quantity sample after training with the small subsample. We are not sure why these two dogs were able to spontaneously generalize in the first trial when most of the other dogs (4/5) did not. One of the deployed dogs false alerted in most of the experiment, thus, generalization to the large-quantity sample could have been a fortuitus event (33% chance of identifying the correct bag by chance), which was reinforced. Nevertheless, this could also suggest that in addition to training, repetitive presentation to the large-quantity sample could have incidentally improved generalization for these two dogs. However, further research is needed to confirm this hypothesis. To date, it is not clear why some dogs generalize more so than others under these scenarios. Future investigations in genetic, olfactory, or behavioral markers associated with enhanced generalization could lead to an improvement in the selection and training of detection dogs.

There are several important limitations to this study. First, the changes in odor concentration associated with changes in quantity are not directly tested or known in this case. Although such large changes in quantity would presumably have changes in available odor concentration, it is important to note that factors such as surface area and odor restriction of the bag were not controlled or assessed. This was due to the substantial logistical and security challenges necessary to complete the trials reported, but this does limit our ability to translate quantity changes to concentration changes. Further, a sample of 7 dogs and only 1–4 trials per dog for each experiment is small. Importantly, however, clear trends did emerge and are important to report this opportunistic study result.

Nonetheless, these findings have important implications to detection dog teams and trainers. First, our findings show that the inability of dogs to generalize to different quantities of an odorant in the laboratory also occurs in operational settings, and it is not just a laboratory phenomenon. This generalization deficit represents a major challenge to detection dogs and could be detrimental to public safety. Second, our data suggest that dogs can be trained to detect high concentrations of an explosive. Thus, to mitigate the negative effect of this generalization phenomena, dogs must be trained to alert to different concentration/quantity of a target odor. Although training with high quantities of the target odor could be implemented with little risk for some detection dogs (e.g., narcotics detection dog), the implementation of this training method might be particularly challenging or dangerous for explosive detection dogs. Most of the research in canine generalization has evaluated training methods to improve generalization to different variants or mixtures of the target odor [5,12,14,15,16,17,18,19,28]. However, to our knowledge, no other study has evaluated generalization to different quantities of an explosive. Additional research should be conducted to develop and evaluate different training methods that could help mitigate concentration generalization problems in explosive detection dogs, but that at the same time are practical and safe to implement.

## 4. Conclusions

The present study evaluated why two certified detection dogs failed to detect 13 kg of explosive. Results from our first experiment suggest that this lack of response was not due to a context effect as all dogs detected the frequently used training sample when tested in a similar context. Most of the dogs were able to generalize to a 30 g subsample of the confiscated explosive, but with some decrement compared to the typical training sample. However, most dogs did not generalize to the 13 kg of the confiscated ANFO after training with a 30 g subsample of the same explosive. Altogether, these results suggest that dogs may not spontaneously generalize to high quantities of an explosive when only trained with small quantities but can learn to alert to high quantities with explicit training. Further research must be conducted to evaluate this generalization phenomena and develop training methods to mitigate this problem that could represent a threat to public safety.

## Figures and Tables

**Figure 1 animals-11-01341-f001:**
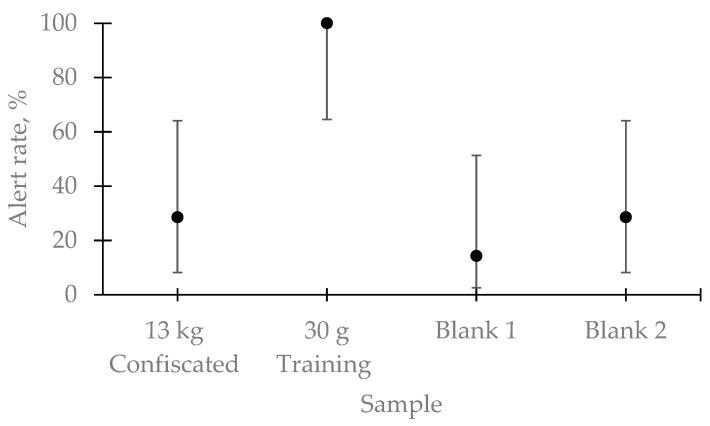
Detection dog teams (*N* = 7) mean alert rate ± Wilson’s score 95% confidence interval to the different samples (bags) tested. The effect of sample was statistically significant (*p* = 0.002). All dog teams alerted to the bag containing the training sample, but not to the 13 kg explosive of the incident.

**Figure 2 animals-11-01341-f002:**
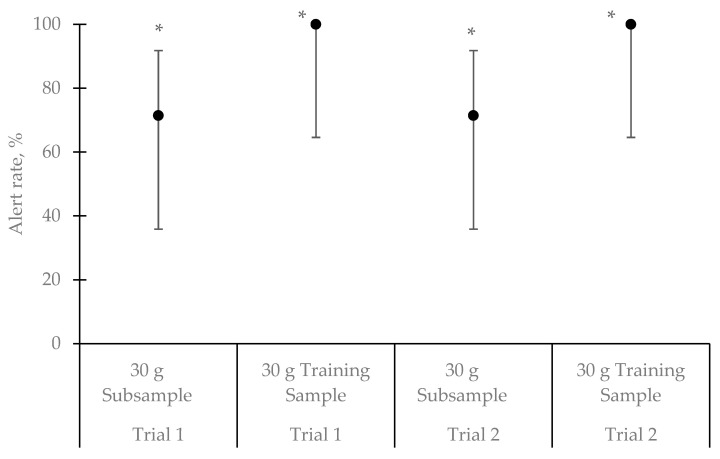
Detection dog teams (*n* = 7) mean alert rate ± Wilson’s score 95% confidence interval to the different samples (confiscated and training) during both trials of the generalization test. The effect of trial, sample, and their interaction were not statistically significant. The alert rates of samples with * were significantly greater than chance. Above chance was based on a binomial test where the probability of a correct response was set to 0.20 (1 out of 5 chance given a five-choice line up). Four of seven dogs making a correct response on a trial would indicate detection greater than chance (0.20). Dog teams alerted to the 30 g of the unknown sample and to the training sample at a rate greater than chance in both trials, but not all dogs were able to generalize to the 30 g of the unknown sample.

**Figure 3 animals-11-01341-f003:**
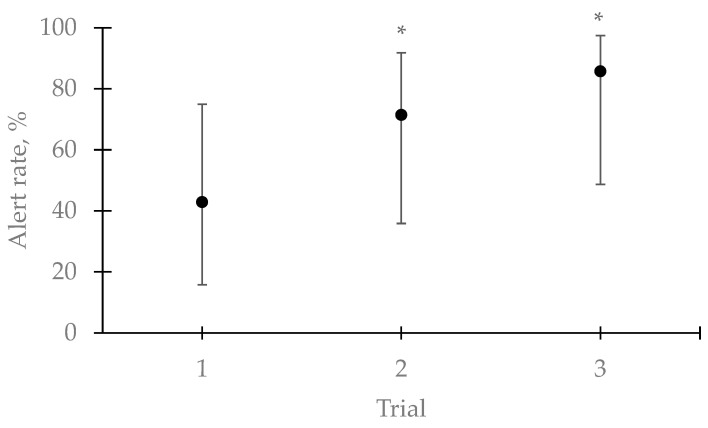
Detection dog teams (*n* = 7) mean alert rate ± Wilson’s score 95% confidence interval to 13 kg of the confiscated ANFO after been trained with 30 g of the same sample. The effect of trial did not quite reach statistical significance (*p* =0.10). The alert rates of trials with * were significantly greater than chance based on a binomial test where the probability of correct responses was set at 0.33. Most dog teams did not generalize spontaneously to the 13 kg sample during the first trial. However, with subsequent training, the alert rate increased in the following trials to levels above chance.

**Table 1 animals-11-01341-t001:** Information about the canine and detection team.

Team	Age, Years	Breed	Sex *	Years of Service
1	4	Labrador	M	3
2	5	Labrador	M	4
3	9	Labrador	F	8
4	2	German Wirehaired Pointer	M	1
5	6	Labrador	M	5
6	6	Labrador	M	5
7	4	Labrador	M	3

* Male and female dogs were neutered/spayed.

## Data Availability

All data are provided as supplementary materials of this article.

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
