# Peer review of "Case Study: An Evaluation of Detection Dog Generalization to a Large Quantity of an Unknown Explosive in the Field"

_animals, 2021, doi:10.3390/ani11051341_

Round 1
Reviewer 1 Report
The authors opportunistically followed up on a “real-life” event in which two highly trained detection dogs failed to detect a large quantity of explosives in a field by organizing three experiments with 7 dog teams: 1. Confirm the scenario and examine role of context; 2. Generalization test; 3. Effect of large quantities of explosives on generalization. The results suggest that dogs should be trained with large quantities of explosives because they typically do not generalize from training with smaller quantities. This finding is interesting and important because it has implications for canine training techniques and may inspire future research. However, the paper should be carefully checked for language and style; there are issues throughout most sections of the main body of the paper. My specific comments are detailed below; they are largely editorial in nature.
Title:
Most readers will have no idea what “ANFO” stands for; I would replace with “explosive”.
Abstract:
Line 24: Provide more detail on what is meant by “training sample”.
Keywords: Include explosive as a keyword; write out ANFO, if you keep the abbreviation as a keyword; delete semi-colon at end of list.
Introduction:
Line 57: Change to “improvised” and “built”
Line 60: Change to “different from”
Line 62: Change to “on this topic”
Line 63: Change to “suggests that canines”
Line 80: Change to “10 fold”
Line 81: Change to “in odor detection thresholds”
Line 86: Sentence beginning “Two of a seven” needs to be rewritten.
Line 87: Change to “in an open field”
Materials and methods:
General comment: I would consider the research as a single study with three experiments; as currently written, the terms switch back and forth (e.g., Section 2.2 is Experiment 1; Section 2.3 is Study 2). This will need to be fixed throughout the paper.
General comment: I am not sure that Animals allows this, but some journals allow authors with experiments that build upon one another to organize the text as follows (after the general methods section): Experiment 1 Methods; Experiment 1 Results; Experiment 2 Methods; Experiment 2 Results; Experiment 3 Methods; Experiment 3 Results. If this is possible, then you would not have to review results of Experiment 1 at the start of methods for Experiment 2 (lines 149-153) and results of Experiment 2 at the start of methods for Experiment 3 (lines 177-180). Alternatively, you could avoid detailed descriptions of results in the methods and simply state, “Based on results from Experiment 1, we conducted Experiment 2, a generalization test.”
Line 109: “Data” is plural, so “Data were”
Line 113: After “observational study”, I would add “, not requiring an animal care and use protocol”, if that was the case.
Line 115: Table 1 footnote: “neutered”
Line 133: “unknown sample”
Line 141: Provide a brief description of the trained behavioral response associated with an alert.
Line 153: “reason for why”
Line 154-156 repeats information provided in the Introduction.
Line 166: “unknown sample”
Line 193: “between trials”
Line 196: “detection of”
Line 206: “Statistical analyses”
Results:
Include X and Y axes in Figures 1-3?
Line 272: Delete extra period.
Discussion:
Line 319: “perception of the ANFO”
Lines 340-348: The authors do a nice job summarizing study limitations.
Author Response
Thank you for your comments and for helping us making a better manuscript. Please see the attachment.

Reviewer 2 Report
I have attached a file.

Author Response
Thank you for reviewing our manuscript and helping us to make it better. Please see the attachment with our response to your comments.

Reviewer 3 Report
See the joined file

Author Response
Thank you for reviewing our manuscript. Please see the attachment with our response.

Round 2
Reviewer 1 Report
I reviewed an earlier version of this manuscript and find the revision much improved with regard to organization (specifically in presentation of methods and results of each experiment) and clarity. All of the typographical errors that I noted in the earlier version have been corrected but I found a few more (see below).
Line 124: Change "were to "was"
Line 135: Change "Cochran tests" to "Cochran test"
Line 202: Change "experiment 1" to "Experiment 1"
Line 233: Change "lineal model" to "linear model"
Line 259: Change "noticeable" to "noticeably"
Line 380: Change "trial one" to "trial 1" to be consistent with naming of the other trials in the sentence
Line 453: Change "folds" to "fold"
Line 477: Change "decreased" to "decrease"
Author Response
All the grammar errors were corrected in the manuscript. Thanks for your revisions.
Reviewer 3 Report
included in the joined file
